# Insulin-Degrading Enzyme Interacts with Mitochondrial Ribosomes and Respiratory Chain Proteins

**DOI:** 10.3390/biom13060890

**Published:** 2023-05-26

**Authors:** Ayse Yilmaz, Chiara Guerrera, Emmanuelle Waeckel-Énée, Joanna Lipecka, Barbara Bertocci, Peter van Endert

**Affiliations:** 1Institut Necker Enfants Malades, Université Paris Cité, INSERM, CNRS, F-75015 Paris, France; ayse.yilmaz@inserm.fr (A.Y.); emmanuelle.enee@inserm.fr (E.W.-É.); barbara.bertocci@inserm.fr (B.B.); 2Structure Fédérative de Recherche Necker, Proteomics Platform, Université Paris Cité, INSERM, CNRS, F-75015 Paris, France; chiara.guerrera@inserm.fr (C.G.); joanna.lipecka@inserm.fr (J.L.); 3Service Immunologie Biologique, AP-HP, Hôpital Universitaire Necker-Enfants Malades, F-75015 Paris, France

**Keywords:** mitochondrion, respiratory chain, mitochondrial translation, chaperone

## Abstract

Insulin-degrading enzyme (IDE) is a highly conserved metalloprotease that is mainly localized in the cytosol. Although IDE can degrade insulin and some other low molecular weight substrates efficiently, its ubiquitous expression suggests additional functions supported by experimental findings, such as a role in stress responses and cellular protein homeostasis. The translation of a long full-length IDE transcript has been reported to result in targeting to mitochondria, but the role of IDE in this compartment is unknown. To obtain initial leads on the function of IDE in mitochondria, we used a proximity biotinylation approach to identify proteins interacting with wild-type and protease-dead IDE targeted to the mitochondrial matrix. We find that IDE interacts with multiple mitochondrial ribosomal proteins as well as with proteins involved in the synthesis and assembly of mitochondrial complex I and IV. The mitochondrial interactomes of wild type and mutant IDE are highly similar and do not reveal any likely proteolytic IDE substrates. We speculate that IDE could adopt similar additional non-proteolytic functions in mitochondria as in the cytosol, acting as a chaperone and contributing to protein homeostasis and stress responses.

## 1. Introduction

IDE is a ubiquitous 110 kDa spherical protease with poorly understood functions that is found in both extra- and intracellular compartments, including the cytosol and, also, to a lesser extent, in the plasma membrane, endosomes and peroxisomes. The protein sequence of IDE is strikingly conserved amongst distantly related species, including those that do not produce insulin. Identity scores between sequences from human, chimpanzee, rat, and mouse range between 95% and 99%. In other vertebrates, such as the zebrafish, IDE is still 85% identical to the human sequence. These high identity scores suggest a common evolution, similar 3D structure and functions, and potentially important yet to be identified roles in the biology of the cell. IDE degrades insulin with high efficacy, but also degrades several amyloidogenic peptides, such as amyloid-beta, IGF-II, glucagon, and amylin [1] Ubiquitin and particularly its less stable forms have also been shown to be degraded by IDE, although ubiquitin can act as an IDE inhibitor, reducing insulin degradation [2,3,4]. Structural analysis by the Tang laboratory has revealed how IDE recognizes and cleaves conformationally metastable amyloidogenic peptides [5].

IDE is composed of two homologous domains connected by a short hinge, one of which contains the zinc ion binding HXXEH motif [6]. IDE can adopt an open conformation, giving substrates access to an inner chamber in which the active site is reconstituted upon closure [5]. Substrates are unfolded inside the inner chamber, which selects substrates of <10 kDa [7]. IDE has an exosite interacting with the N-terminus of substrates and facilitates their unfolding, as well as an allosteric site that regulates the degradation of short peptides [8,9,10].

In humans and rodents, genetic data and deletion experiments suggest a role of IDE in glucose homeostasis: IDE gene knockout in mice results in glucose intolerance, though the mice remain normoglycemic and are only slightly hyperinsulinemic [11]. In addition, the IDE gene is linked to T2D and Alzheimer’s disease in humans. Acute treatment with two structurally different inhibitors designed independently by the teams of Deprez-Poulain in collaboration with us and Maianti produced only partially consistent effects on glucose tolerance in mice, underlining the complexity of the interaction between IDE and glucose and insulin homeostasis in vivo [12,13].

The reason underlying the high level of sequence conservation across species, and the ubiquitous expression of IDE, including in cells lacking identified IDE substrates, remains unknown. However, over the last decades, experimental findings have established that IDE is also involved in a wide variety of biological processes. Tundo et al. demonstrated that normal and malignant cells exposed to different stresses up-regulate IDE in a heat shock protein-like fashion, and proposed that IDE binds proteins as a “dead end chaperone” [14]. Consistent with this, IDE was initially reported to function as receptor for the VZV virus but was then shown to bind to an intracellular VZV protein, most likely a misfolded fragment [15,16]. Literature reports have also documented that IDE co-immunoprecipitates with the proteasome and ubiquitin [17,18]. In line with a possible role in protein scavenging following ER stress, Schmitz et al. reported an IDE-dependent clearance mechanism for ER-localized amyloid-β, suggesting that IDE may be involved in a parallel ERAD pathway that is not dependent on the proteasome [19]. We have recently found that IDE deficiency triggers the unfolded protein response that is enhanced upon metabolic or pharmacologic stress and is accompanied, at least in pancreatic islet cells, by cell proliferation (Zhu et al., manuscript submitted). Thus, the available data suggest that IDE plays a role in protein homeostasis, particularly in stress situations as well as in glucose metabolism. However, the mechanistic basis for these roles remains unclear.

Although wild-type IDE is predominantly located in the cytosol in all cell types studied so far, IDE is also found in the mitochondria. The amino acid sequences of full-length human and mouse IDE features two alternative in-frame initiating methionines at the positions 1 (Met^1^) and 42 (Met^42^). A bioinformatics analysis of the full-length IDE protein sequence predicts the presence of a mitochondrial targeting sequence (MTS) downstream of Met^1^, a prediction confirmed by the mitochondrial localization of eGFP preceded by the N-terminal portion of Met^1^-IDE [20]. Importantly, although the presence of a more efficient Kozak sequence promotes the dominant translation initiation at Met^42^ resulting in dominant cytosolic localization, a fraction of IDE can be detected in mitochondria from wild-type CHO and HEK cells, indicating the physiologic relevance of translation initiation at Met^1^. Observing highly efficient in vitro cleavage of a synthetic MTS by IDE, possibly related to the frequent presence of arginine residues preferred by IDE in MTS, Leissring and colleagues speculated that IDE might contribute to the physiologic cleavage of MTS assisting other critical enzymes [21,22]. However, mitochondrial IDE might also have more important roles. For example, IDE has been suggested to cooperate with the mitochondrial SIRT4 deacetylase to mediate the lysosomal degradation of PTENα and promote survival in response to starvation stress [23]. Due to its capacity of cleaving amyloid-β, IDE might also help prevent the accumulation of toxic Ab in mitochondria [24]. Considering the complete lack of information about the biological roles of IDE in mitochondria, we set out to identify interactions of IDE with mitochondrial proteins using a proximity biotinylation approach. We report that both wild-type and protease-dead mitochondrial IDE interact primarily with proteins involved in mitochondrial protein translation and in the citric acid cycle. While the functional impact of these interactions remains to be determined, we speculate that mitochondrial IDE, like the dominant cytosolic form, might be involved in mitochondrial protein homeostasis, particularly in stress situations.

## 2. Materials and Methods

### 2.1. Generation of TurboID (Biotin Ligase BirA Mutant) Fusion Constructs, Lentivirus Production and Transduction

Lentiviral constructs were built in the vector pLVX-tdTomato-N1 (Takara, Kyoto, Japan), which was first modified by the insertion between XhoI (5′) and BamHI (3′) in the multiple cloning site of a synthetic gene, containing (5′ to 3′) SmaI and MluI restriction sites, the sequence coding for P2A/T2A self-cleaving peptides (ATNFSLLKQAGDVEENPGP/EGRGSLLTCGDVEENPGP), and td-Tomato, resulting in the vector pLVX-P2A-T2A-tdTomato.

To generate the lentiviral expression constructs IDE-TurboID and IDEMut E111D-TurboID, full-length cDNA (complementary DNA) encoding wt or protease-dead human IDE was amplified from the previously published constructs IDE- or IDEMut E111D-pCRBlunt [25], with primers adding XhoI at the 5’ end and including an EcoRI site ahead of the stop codon. The PCR product was first cloned as an XhoI-EcoRI fragment into the vector pMA-RQ previously modified by the insertion of a synthetic sequence comprising 79 base pairs at the 3′ end of IDE, including the EcoRI site but not the stop codon, and sequences encoding a GSAGSA linker, the V5 peptide, the TurboID enzyme, and a MluI site at the 3′ end. The sequence encoding TurboID preceded by the V5 peptide was taken from Branon et al. [26]. Finally, the complete constructs were transferred as XhoI-MluI fragments into pLVX-P2A-T2A-tdTomato.

The construct pLVX-MTS-OVA-V5-TurboID-td-Tomato was produced by amplifying a fragment comprising the GSAGSA linker, V5, and TurboID DNA sequences with primers containing SmaI (5′) and MluI (3′) restriction sites and inserting the PCR product into the plasmid pLVX-MTS-OVA-T2A-P2A-tdTomato previously generated in the laboratory. The latter plasmid carries the mitochondrial targeting sequence (MTS) from human cytochrome c oxidase (MSVLTPLLLRGLTGSARRLPVPRAKIHSL; NCBI, NM_205152.3) and SmaI and MluI sites between the T2A-P2A and tdTomato sequences. Synthetic genes were purchased from GeneArt (Thermo Fisher Scientific, Waltham, MA, USA). The primers for PCR amplification are listed in Appendix A.

Plasmid DNA was extracted and purified with the Nucleobond Xtra Midi EF kit (Macherey-Nagel, Düren, Germany) and used to produce lentiviruses with an average titer of 10^9^ TU/mL. The HEK (human embryonic kidney) 293 cell line was transduced at a multiplicity of infection (MOI) of 5 for 6 h. Cells were selected starting 48 h after infection with puromycin at the concentration of 10 μg/mL. After 2–3 passages in selection media, the percentage of transduced cells was determined by quantifying td-Tomato positive cells by flow cytometry.

### 2.2. Confocal Microscopy

4 × 10^4^ cells were seeded onto glass coverslips pre-coated with Poly-D-Lysine overnight. The cells were fixed for 12 min with a solution containing 4% paraformaldehyde, 0.2% glutaraldehyde, 60 mM PIPES (piperazine-*N,N′*-bis(2-ethanesulfonic acid)), 25 mM HEPES((4-(2-hydroxyethyl)-1-piperazineethanesulfonic acid), 10 mM EGTA (ethylene glycol-bis(β-aminoethyl ether)-*N,N,N′,N′*-tetraacetic acid), 2 mM Magnesium acetate, and then permeabilized with 0.2%Triton X-100 in DPBS (Dulbecco’s phosphate-buffered saline) for 10 min. After blocking with 5% donkey serum in PBS-0.05% Tween-20 for 2 h at RT (room temperature), the cells were incubated sequentially with primary antibodies overnight at 4 °C (IDE, V5, TOM20 all at 1:100), and with the appropriate secondary antibodies (1:200) for 2 h at RT (Appendix A). Nuclear counterstaining was carried out using DAPI (4′,6-diamidino-2-phenylindole) at the concentration of 1 μg/mL. The slides were mounted with Vectashield Plus Antifade media. The image acquisitions were performed with a 63x oil immersion objective (NA 1.4) and a laser scanning confocal microscope (TCS SP8-3X STED; Leica Microsystems, Weztlar, Germany). The images were processed with Icy (https://icy.bioimageanalysis.org, accessed on 9 May 2022).

### 2.3. Western Blot

Standard denaturing SDS-PAGE analysis was performed using Mini Gel Tank equipment (Thermo Fisher Scientific, Waltham, MA, USA). Proteins were transferred onto PVDF (polyvinylidene difluoride) membrane with the iBlot 2 Dry Blotting System (Thermo Fisher Scientific). The membranes were blocked with BSA (Bovine serum albumin) 5% in TBS (Tris-buffered saline)-Tween 0.1% 1 h at RT and then incubated with the primary antibody overnight at 4 °C and with the secondary antibody 2 h at RT. Antibody binding was visualized by enhanced chemiluminescence with SuperSignalTM West Pico PLUS Chemiluminescent Substrate (Thermo Fisher Scientific), and the images were acquired on the ChemiDoc Imaging System (BioRad, Hercules, CA, USA). Primary and secondary antibodies are listed in the Supplemental Materials (Appendix A).

### 2.4. TurboID-Based Enzymatic Protein Labeling and Extraction of Biotinylated Proteins for Proteomic Analysis

Proximity labeling by the TurboID enzyme fused to IDE and MutIDE expressed in HEK cells was performed according to the method described by Cho and coworkers [27]. The TurboID system uses a fast biotin ligase so that incubations of a few minutes are usually sufficient for the biotinylation of neighboring proteins. Briefly, 20 × 10^6^ HEK293T cells expressing the IDE-TurboID proteins were cultured at 37 °C 5% CO_2_ in Dulbecco’s Modified Eagle Medium (DMEM) containing 10 U/mL penicillin-streptomycin, 2 mM Glutamine (Sigma-Aldrich, St. Louis, MO, USA), and 10% dialyzed FCS (Fetal calf serum, Eurobio Scientific, Saclay, France) for 24 h. The cells were then incubated with 50 μM biotin (Sigma-Aldrich) for 15 min. The labeling reaction was stopped by placing the cells on ice and washing out the biotin excess with ice-cold DPBS (Thermo Fischer Scientific). The cells were detached via pipetting with 10 mL of ice-cold DPBS and centrifuged at 300× *g*. The cell pellet was lysed in 2 mL of a radio immunoprecipitation assay (RIPA) lysis buffer (Thermo Fisher Scientific) supplemented by 1x protease inhibitor cocktail (Complete EDTA (Ethylenediaminetetraacetic acid)-free protease inhibitors, Roche Diagnostic, Basel, Switzerland) and incubated on ice for 30 min. The cell lysate was clarified by centrifugation at 13,000× *g* at 4 °C for 10 min and transferred to a fresh tube. The protein concentration was measured by a BCA (bicinchoninic acid) protein assay (Bio-Rad, Hercules, CA, USA). For the enrichment of the biotinylated proteins, 2.5 mg proteins in a 1.8 mL RIPA buffer were incubated with 200 μL Streptavidin magnetic beads (Thermo Fisher Scientific) overnight at 4 °C. Afterward, the supernatant was removed using a magnetic rack to pellet the beads, which were washed twice with the RIPA buffer (1 mL, 2 min), once with KCl 1 M (1 mL, 2 min), once with Na_2_CO_3_ 0.1 M (1 mL, 10 s), and once with urea 2 M in 10 mM Tris-HCl (pH 8.0) (1 mL, 10 s). After the final wash, the beads were transferred in a 1 mL RIPA buffer to a fresh tube and washed again with a RIPA buffer (1 mL, 2 min). The enriched biotinylated proteins were eluted from the beads in 80 μL of 4× Laemmli buffer (Biorad) supplemented with 2 mM biotin and 20 mM DDT (dichloro-diphenyl-trichloroethane) at 95 °C for 10 min. The input, the flow-through, and the enriched material from each sample were analyzed by western blot before performing a proteomic analysis.

### 2.5. Identification and Quantification of Proteins by NanoLC-MS/MS

Sample digestion was performed on S-TrapTM microcolumns (Protifi, Hutington, CA, USA) according to the manufacturer’s instructions. Samples (60 µL, IP) were supplemented with 20% SDS to a final concentration of 5%, reduced with 20 mM TCEP, and alkylated with 50 mM CAA (chloracetamide) for 15 min at room temperature. Aqueous 27.5% phosphoric acid was then added to a final concentration of 2.5%, followed by the addition of a binding buffer (90% methanol, 100 mM TEAB, pH 8). The mixtures were then loaded onto S-Trap columns and centrifuged for 30 s at 3000× *g*. The columns were washed 6 times (3000× *g*, 30 s) to ensure the complete removal of SDS. The samples were digested with 1 µg trypsin (Promega, Madison, WI, USA) at 47 °C for 1.5 h. After elution, the peptides were dried under a vacuum and resuspended in 20 µL of 2% ACN and 0.1% formic acid in HPLC grade water prior to MS analysis.

An amount of 2 µL of the tryptic peptides were injected onto a nanoElute HPLC system (Bruker Daltonics, Leipzig, Germany) coupled to a timsTOF Pro mass spectrometer (Bruker Daltonics). HPLC separation (Solvent A: 0.1% formic acid in water, 2% acetonitrile; Solvent B: 0.1% formic acid in acetonitrile) was performed at 250 nL/min using a packed emitter column (C18, 25 cm × 75 μm 1.6 μm) (Ion Optics, Fitzroy, Australia) using gradient elution (2–11% solvent B for 19 min; 11–16% for 7 min; 16–25% for 4 min; 25–80% for 3 min and, finally, 80% for 7 min to wash the column). The data were acquired using the PASEF (parallel accumulation serial fragmentation) acquisition method. Measurements were performed over the *m*/*z* range from 100 to 1700 Th. The range of ion mobility values was from 0.85 to 1.3 V s/cm^2^ (1/K0). The total cycle time was set to 1.2 s, and the number of PASEF MS/MS scans was set to 6.

### 2.6. Data Processing after LC-MS/MS Acquisition

The raw file folders (.d) were processed with MaxQuant 2.0.1 software and searched with the Andromeda search engine in the UniProtKB/Swiss-Prot Homo Sapiens database (version 01-02-2021, 20,396 entries). To search for parent and fragment mass ions, we set an initial mass deviation of 4.5 ppm and 20 ppm, respectively. The minimum peptide length was set at seven amino acids, and strict specificity for the trypsin cleavage was required, allowing up to two missed cleavage sites. Carbamidomethylation (Cys) was defined as a fixed modification, while oxidation (Met) and N-term acetylation were defined as variable modifications. Matching between the series was not allowed. The minimum number of LFQ (Label Free quantification) ratios was set at 2. False discovery rates (FDR) at the protein and peptide level were set at 1%. Scores were calculated in MaxQuant as described previously [28]. The reverse and common contaminants were removed from MaxQuant output. Proteins were quantified according to the MaxQuant label-free algorithm using LFQ intensities. Finally, a match between runs was allowed during the analysis.

Four independent immunoprecipitations per group were analyzed with Perseus software (version 1.6.15.0), which is freely available at www.perseus-framework.org, accessed on 16 March 2022 [29]. Protein intensities were log2 transformed, and proteins identified in at least four replicates in at least one group were statistically tested (volcano plot, FDR = 0.05 and S0 = 0.1) after the imputation of the missing value by a Gaussian random number distribution with a standard deviation of 30% from the standard deviation of the measured values and a downward shift of 1.8 standard deviations from the mean. To increase the selectivity of the candidate, a double comparison was performed between the IDE/mutIDE (with biotin) vs the negative control (with biotin), and IDE/mutIDE (with biotin) vs. IDE/mutIDE (without biotin).

### 2.7. IDE Interactome Data Analysis

Functional protein networks of the IDE and MutIDE interactome data were annotated with STRING Software (version 11.5) [30]. Lists of (i) all proteins significantly enriched in IDE or MutIDE samples incubated with biotin relative to both control samples (samples without biotin and OVA samples), and of proteins; (ii) enriched in biotin samples relative to OVA controls only with fold change > 3.5; or (iii) enriched in biotin samples relative to samples without biotin only with fold change > 5.5, were inputted into the String database in the section “multiple protein analysis”. The organism was set as Homo Sapiens. The parameter chosen for network analysis was “full string network” with a minimum required interaction score of 0.04. Interactions were limited to protein query only. The PPI (Protein Protein interaction) enrichment *p*-value was ≤10^−16^ and the average local clustering coefficient varied between 0.5–0.6 for the data sets analyzed.

The pathway enrichment analysis was analyzed with Metascape (https://metascape.org, accessed on 10 March 2023) [31]. The list of selected genes was submitted to Metascape (Version v3.5.20230101). The process enrichment analysis was carried out with the following ontology sources: KEGG Pathway, GO Biological Processes, Reactome Gene Sets, Canonical Pathways, CORUM, and WikiPathways. The complete human genome was used as the enrichment background. Terms with a *p*-value < 0.01, a minimum count of 3, and an enrichment factor > 1.5 (i.e., the ratio between the counts observed and the counts expected by chance) were collected and grouped into clusters based on their membership similarities. *p*-values are calculated based on the cumulative hypergeometric distribution, and q-values are calculated using the Benjamini-Hochberg procedure to account for multiple testing. The most statistically significant term within a term cluster is chosen to represent the cluster. Top clusters with their representative enriched terms (one per cluster) are depicted as bar graphs with a discrete color scale indicating statistical significance expressed as −log10.

## 3. Results

### 3.1. Design and Verification of the Proximity Biotinylation System

#### 3.1.1. Constructs and Protein Expression

To identify proteins interacting with mitochondrial IDE, we set up a proximity biotinylation system. We constructed a fusion protein in which a full-length IDE sequence starting at Met^1^ was joined via a V5 tag to the biotin ligase (Figure 1A). To allow for the flow cytometric selection of cells expressing the construct, a cDNA encoding fluorescent tdTomato was joined to the ligase C-terminus, separated by a self-cleaving peptide. Considering that some functions of IDE may be independent of its proteolytic activity, notably in the response to proteotoxic stress [25], we also expressed mutant IDE with a Glu-Asp mutation at position 111, which decreased the enzyme activity to <1% [32]. To produce a control TurboID fusion protein residing in the mitochondria, we designed an analogous construct in which IDE was replaced by OVA, preceded by an efficient mitochondrial targeting sequence from human cytochrome C oxidase. All fusion constructs were inserted into a lentiviral expression vector and used to transduce human embryonal kidney cells (HEK293). After puromycin selection, >80% of the cells were tdTomato positive in the flow cytometric analysis. TdTomato fluorescence was then used routinely to confirm the continued high-level expression of the fusion constructs.

To verify the fusion protein expression, we analyzed the cell lysates by immunoblot (Figure 1B). Cells transduced with viruses encoding IDE displayed a band with the molecular weight expected for the fusion of IDE with TurboID (162 kDa), in addition to the endogenous IDE protein (118 kDa), and a minor band likely corresponding to the entire fusion protein including tdTomato (215 kDa). Consistent with this, major and minor bands corresponding to cleaved and uncleaved fusion proteins, respectively, were detected using an antibody to the V5 tag. An additional protein migrating just above the endogenous IDE protein and reacting with IDE and V5 antibodies may correspond to aberrant cleavage just downstream of the V5 tag. An analysis of cells expressing OVA revealed bands corresponding to the expected cleaved fusion protein (85 kDa) and a band (about 47 kDa) that may again correspond to cleavage just after the V5 tag.

#### 3.1.2. Protein Localization

To verify the intracellular localization of the TurboID fusion protein, we analyzed HEK cells transduced with lentiviruses encoding the IDE and OVA fusion proteins by confocal microscopy, using antibodies to the two proteins and to the V5 tag as well as an antibody recognizing TOM20, a subunit of the translocase of the mitochondrial outer membrane. As expected, TOM20 marked the contours of mitochondria in all cells (Figure 2A–D). IDE staining of control HEK cells revealed a diffuse cytosolic pattern consistent with the dominant expression of the Met^42^ IDE transcript that results in cytosolic IDE localization (Figure 2C). In contrast, the staining of both wt and protease-dead IDE, as well as mitochondrial targeted OVA, revealed exclusive vesicular staining that was consistent with mitochondria. In merged images, IDE and OVA antibody-stained structures appeared encircled by TOM20 staining, suggesting that both proteins localized to the internal mitochondrial space and not to the outer membrane.

#### 3.1.3. Biotinylation Setup

We then analyzed the efficacy of protein biotinylation by the TurboID fusion proteins, using the immunoblot staining of bulk biotinylated proteins (Appendix A). The immunoblot analysis revealed several background bands evident in control and transduced cells and unaffected by biotin addition. In the transduced cells, the biotin addition resulted in the appearance of a major band likely corresponding to the auto-biotinylation of the cleaved IDE fusion protein (162 kDa), as well as a series of faint bands migrating between markers of 15 to 70 kDa. The FCS dialysis had little effect on the background but reduced fusion protein auto-biotinylation. There were no biotin dose effects, so we used standard FCS and 50 μM biotin in all subsequent experiments.

To perform the preparative scale experiments, we applied these conditions to batches of 20 × 10^6^ cells. An immunoblot analysis of extracts and eluted proteins from cells expressing IDE or OVA fusion proteins not incubated with biotin revealed a dominant band around 70 kDa that was moderately enriched in eluates (Appendix A). In contrast, the addition of biotin resulted in the appearance of multiple bands that were strongly enriched by affinity purification. Curiously, and in contrast to preliminary analytical results, affinity purification enriched most strongly for auto-biotinylated OVA, but not IDE. Thus, the analysis of preparative scale experiments showed the strong biotin-dependent enrichment of proteins binding to streptavidin, so we proceeded to the proteomic analysis.

### 3.2. Identification of Proteins Interacting with Wt and Mutant Mitochondrial IDE

The proteomic analysis of samples from independent replicates of cells expressing IDE fusion proteins identified a total of almost 4000 proteins. To identify proteins that were enriched significantly, we performed a double comparison of proteins detected in biotin-incubated samples with OVA spatial compartmentalization samples and IDE samples incubated without biotin. Using a fold change (FC) threshold S0 of 0.1 and an FDR threshold of 0.05 (5%), this yielded 62 proteins significantly enriched for wt IDE, and 53 enriched for MutIDE, respectively (Appendix A). Among these, the majority was shared between the wt and mutant IDE (proteins underlined, or neither underlined nor boxed), although fold enrichment varied between the two forms, while some proteins (boxed) were enriched in wt IDE only (Figure 3A).

The molecular weight of the 62 proteins interacting with IDE ranged from 8 to 116 kDa (Figure 3A), with only a single protein of less than 10 kDa (NDUFA1). Thus, considering the length limits for proteins hydrolyzed by IDE [8], all but one protein was unlikely to be an IDE substrate. Consistent with our immunoblot analyses showing strong IDE auto-biotinylation (Appendix A), IDE was most strongly enriched relative to the controls. Among the 62 proteins identified, 47 are known to be localized in mitochondria. The 15 non-mitochondrial proteins include proteins involved in RNA metabolism (e.g., RQCD1, CARS, RAVER1), histone-modifying enzymes (e.g., TRMT112, CBX6, RBP4, PAF1), and various cytosolic proteins. Five of the eight proteins unique to wt IDE samples were non-mitochondrial. Among the mitochondrial proteins, 13 were components of the small or large mitochondrial ribosome subunit. Another nine proteins were subunits of the mitochondrial respiratory chain complex I [6], complex III [2], or complex V [1]. Other mitochondrial proteins were involved in protein synthesis (GADD45GIP1), membrane potential (PPA2), and protein folding and import (HSPE1).

To identify known physical or functional interactions among the proteins identified, we used the STRING database and algorithm [30]. This revealed three functionally associated mitochondrial protein networks (Figure 3B): mitochondrial protein translation with 17 members, of which many are known to interact physically; a citric acid cycle and respiratory electron transport with 15 members, again with most known to interact physically; and fatty acid metabolism (four members). Additional non-mitochondrial networks included the heterocycle metabolic process and networks related to RNA metabolism (tRNA metabolism, CCR4-NOT complex). A complementary functional enrichment analysis using Metascape [31] confirmed the identification of the three mitochondrial networks, showing the highly significant enrichment of mitochondrial gene expression(i.e. ribosomal translation) the TCA cycle and respiratory electron transport (Figure 3C).

We then analyzed the proteins significantly enriched in samples from cells expressing MutIDE in the same manner (Figure 4A–C). Nineteen of 53 proteins (33%; including six of nine proteins unique to MutIDE) were non-mitochondrial and included proteins related to histones, tRNA metabolism and mRNA processing, as for wt IDE, but also two plasma membrane glucose transporters. Again, a single protein (LYRM5, a mitochondrial protein involved in electron transfer) had a Mr of <10 kDa. Thus, under the experimental conditions used, protease-dead IDE does not seem to co-purify with potential proteolytic substrates. An analysis of functional and/or physical protein networks using STRING identified the same three mitochondrial networks (protein translation, TCA cycle, and fatty acid metabolism), which was confirmed by functional enrichment analysis using Metascape. However, we noted a difference between wt IDE and MutIDE regarding non-mitochondrial networks, as the “heterocycle metabolic process” with nine members was present in wt IDE samples only, while “glucose transport” and “mRNA metabolic process” was detected in MutIDE samples only. Given that cytosolic protein networks may reflect the interactions of cytosolic IDE rather than contaminants in our proteomic analyses (see below), these differences may be of interest with respect to the function of the dominant cytosolic IDE form.

### 3.3. Differential Control Analysis Corroborates Mitochondrial IDE Interactome and Reveals Potential Cytosolic IDE Interactomes

The presence of a substantial percentage of significantly enriched cytosolic proteins in both wt and mutant IDE samples (24% and 33%, respectively) was unexpected. Although this could simply represent “background”, this seemed unlikely because many of these cytosolic proteins belonged to protein networks, for example those related to RNA processing. An alternative explanation was a less than perfectly overlapping localization between IDE and our compartmentalization control OVA. Such a difference could result from distinct efficiencies of the MTS used. The MitoFates algorithm for the prediction of mitochondrial pre-sequences attributes an extremely high score of 0.987 to the cytochrome c oxidase MTS preceding OVA, while the complete Met^1^ IDE sequence contains an MTS with the moderate score of 0.399 only [33]; both MTS are predicted to be cleaved by the mitochondrial processing peptidase at positions 11 and 19, respectively. Thus, it was not unlikely that HEK cells transduced with Met^1^ IDE harbored some IDE in the cytosol, while OVA was localized exclusively in the mitochondria. A much stronger expression in the mitochondria could have prevented the detection by microscopy (Figure 2A,B).

To verify this hypothesis, we examined proteins significantly enriched in cells expressing wt IDE compared either to IDE samples without biotin only or to OVA samples only. This showed, first, that the comparison with IDE samples without biotin resulted in stronger enrichment factors for significantly enriched proteins globally, a phenomenon likely due to the background biotinylation of the IDE interactome in OVA cells upon the addition of biotin (Appendix A). In a comparison of IDE samples incubated with versus without biotin, 89 proteins were significantly enriched upon the addition of biotin (Figure 5A). Only five of these were predicted or known to not reside in mitochondria. A STRING analysis identified two mitochondrial protein networks among these: TCA cycle/respiratory electron transport and mitochondrial gene expression, each with numerous members (20 and 33, respectively) (Figure 5A). Neither “fatty acid metabolism” nor any of the three non-mitochondrial protein networks were significantly enriched. A functional enrichment analysis using Metascape again confirmed this analysis, showing the highly significant enrichment of the two networks and of the “mitochondrial RNA metabolic process” in the absence of any non-mitochondrial network (Figure 5B). The examination of the top 20 enriched proteins showed the absence of non-mitochondrial proteins and the strong enrichment of proteins related to mitochondrial translation in both wt and mutant IDE (9 and 11 ribosomal proteins, respectively) and to respiratory electron transport (Appendix A).

We also performed the inverse analysis, comparing the data obtained in biotin-incubated cells expressing wt IDE with cells expressing OVA in mitochondria (Appendix A). Among the 77 proteins significantly enriched in this comparison, only five were mitochondrial. This included two subunits of complex I of the respiratory chain (NDUFA5, NDUFB9) and a large ribosomal subunit (MRPL42) also strongly enriched in the biotin-no biotin comparison. A STRING analysis suggested only two functionally interacting networks: one containing various molecules contained in neutrophile granules, and another one to Parkinson’s disease. However, a Metascape analysis identified a larger number of functional networks in addition to these two which were more strongly enriched, including stress response (15% of members enriched), translation (10% enriched), and RNA metabolism (14% enriched). An examination of the top 20 proteins enriched in the IDE-OVA comparison showed a variety of non-mitochondrial proteins, some of which were associated with the response to oxidative stress (lactoferrin, arginase, several E3 enzymes), and the strongest enrichment for a subunit of V-ATPase. Enrichment factors were lower than in the biotin-no biotin comparison, and only IDE itself reached 10-fold enrichment. Note that IDE was not among the strongly enriched proteins in the biotin-no biotin comparison, suggesting the substantial auto-biotinylation in the absence of added biotin.

## 4. Discussion

In this study, we had set out to obtain initial leads on the role of the fraction of IDE localizing to mitochondria by using a fast proximity biotinylation approach and HEK cells. The expression of a full-length human IDE cDNA starting at Met^1^ resulted in strongly dominant mitochondrial localization, which was consistent with previous reports. The short-term incubation of fusion protein-expressing cells with biotin resulted in the readily detectable biotinylation of client proteins, allowing for the proteomic identification of candidate proteins interacting with IDE. Following standard recommendations, potentially interacting proteins were identified through comparison with a non-biotin and with a compartmentalization control.

Both the confocal microscopy experiments and the results of the proteomics indicate that IDE localizes to the mitochondrial matrix, a conclusion also supported by the analysis of the IDE MTS using the DeepMito algorithm [34]. Given the physiologic localization of cytochrome C oxidase, the same conclusion applied (not surprisingly) to the OVA control. A manual as well as a bioinformatics analysis of the proteomics results strongly suggested that mitochondrial IDE interacts with the mitochondrial protein translation machinery, and with the proteins of the respiratory chain. This was true when both controls were considered but was even more evident when only the no-biotin control was considered. In the latter case, six (wt IDE) and seven (MutIde) of the top ten enriched proteins were ribosomal subunits. Our data also suggest that wt and mutant IDE interact with the same mitochondrial proteins.

We were surprised to find a significant number of non-mitochondrial proteins, some of them linked to protein networks related to RNA processing and stress responses, in the IDE interactome. Given the absence of these proteins when only the no-biotin control was considered, this finding almost certainly resulted from the partial localization of IDE but not OVA fusion proteins in the cytosol due to distinct efficiencies of the IDE and OVA MTS. The relevance of the non-mitochondrial proteins and protein networks as IDE interactants will only become clear upon appropriately controlled investigations of Met^42^-encoded cytosolic IDE. However, proteomic and genomic studies of tissues from IDE-deficient mice in our laboratory are consistent with a role for cytosolic IDE in stress responses and RNA processing (Zhu et al., manuscript submitted).

Non-mitochondrial proteins were not only present in significant numbers in the IDE interactome when both controls were considered, but represented the vast majority of strongly enriched proteins when only the OVA control was considered, a finding requiring explanation. First, due to its strong MTS, OVA very likely was entirely absent from the cytosol, resulting in the extremely low background biotinylation of cytosolic proteins in the OVA sample, with the consequence of relatively strong enrichment factors for proteins biotinylated by cytosolic TurboId-IDE. Second, and because of this, the inspection of the complete list of enriched proteins relative to the OVA control (Appendix A) shows that many ribosomal and other mitochondrial proteins are present, but with lower enrichment factors. Third, the globally lower enrichment factors in the comparison of IDE to OVA controls only suggests that samples incubated with exogenous biotin may display a higher background and therefore a lower enrichment, a factor biasing the detection to cellular compartments devoid of an exogenous biotin ligase (the cytosol in OVA samples).

What might be the role of mitochondrial IDE and its interactions? The fact that almost all proteins biotinylated by both wt and mutant IDE have molecular weights inconsistent with the selectivity of IDE suggests that IDE may have few or no proteolytic substrates in mitochondria. We speculate that IDE may play a role in mitochondrial protein homeostasis and stress responses, possibly as a chaperone-like protein, as has been suggested for the cytosolic protein [14,35]. The impact of IDE on mitochondrial function at the steady state and in stress situations remains to be evaluated.

## Figures and Tables

**Figure 1 biomolecules-13-00890-f001:**
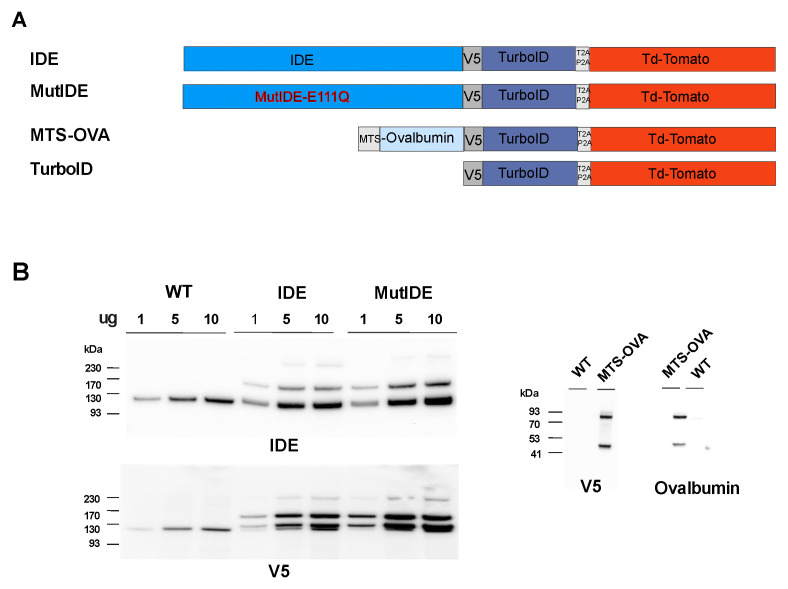
Characterization of TurboID fusion proteins. (**A**) Schematic representation of mitochondria-targeted TurboID constructs for proteomic mapping. Human IDE cDNA or its mutated form (E111Q; MutIDE), starting from Met^1^, and an OVA cDNA were fused to sequences encoding a V5 epitope tag, the TurboID enzyme, the self-cleaving peptide T2AP2A, and the fluorochrome tdTomato; MTS, mitochondrial targeting sequence; (**B**) The expression of TurboID fusion proteins in HEK293 cells. The expression of the IDE and OVA constructs in graded amounts of total protein extracts was analyzed by western blot using antibodies to IDE, OVA and the V5 tag.

**Figure 2 biomolecules-13-00890-f002:**
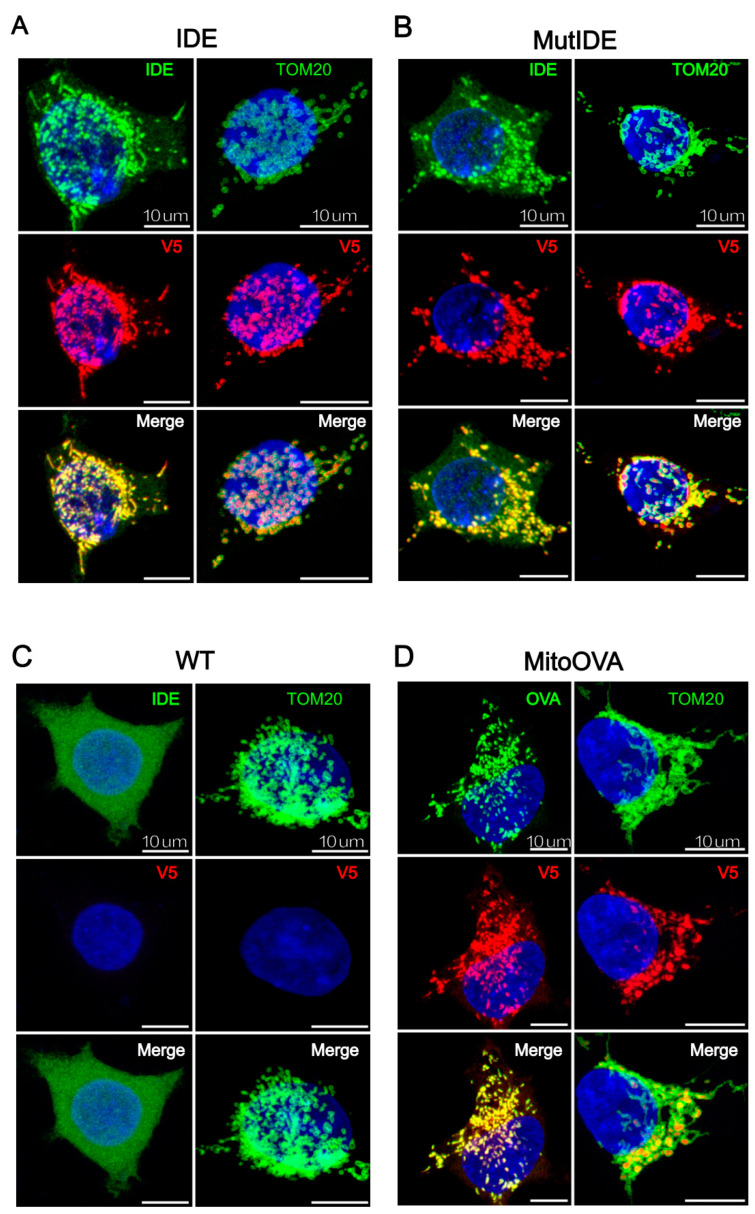
Confocal fluorescent imaging of HEK293 expressing TurboID fusion proteins. Cells expressing IDE-TurboID (**A**); MutIDE-TurboID (**B**); endogenous IDE (**C**); and MTS-OVA-TurboID (**D**) were stained with antibodies to TOM20, IDE, OVA and the V5 tag, as indicated. Merged images show the overlay of TOM20 with fusion protein staining. The nuclei are counterstained with DAPI. Scale bar 10 μm.

**Figure 3 biomolecules-13-00890-f003:**
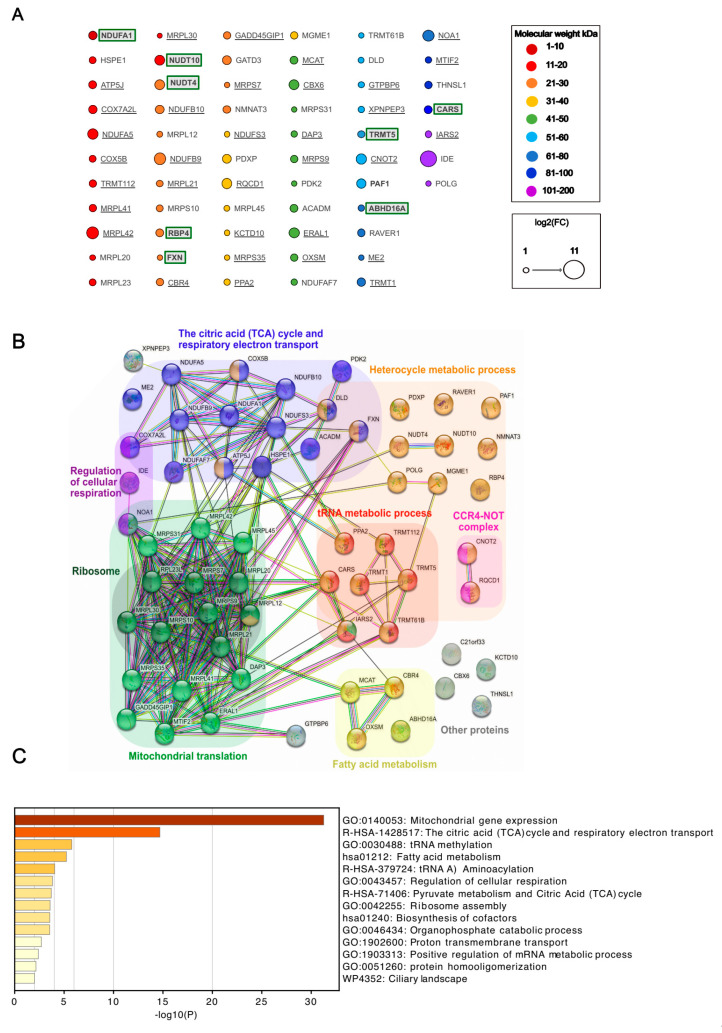
Analysis of the wt IDE interactome. (**A**) Significantly enriched proteins are ranked according to molecular weight (circle colors) and fold change (circle diameter). Proteins shared between the wt IDE and MutIDE interactomes are underlined; proteins restricted to the wt IDE interactome are boxed; proteins neither underlined nor boxed are enriched in the wt IDE interactome relative to both controls and in the mutIDE interactome relative to one control only; (**B**) Functionally associated protein networks in the wt IDE interactome as identified by STRING analysis. Proteins belonging to the same biological system are labeled with an identical color; (**C**) Top non-redundant enrichment clusters in the wt IDE interactome, as identified by Metascape. A total of 32% of genes in GO:0140053 and 19% in R-HSA-1428517 were enriched. The color scale represents statistical significance levels expressed as −log10.

**Figure 4 biomolecules-13-00890-f004:**
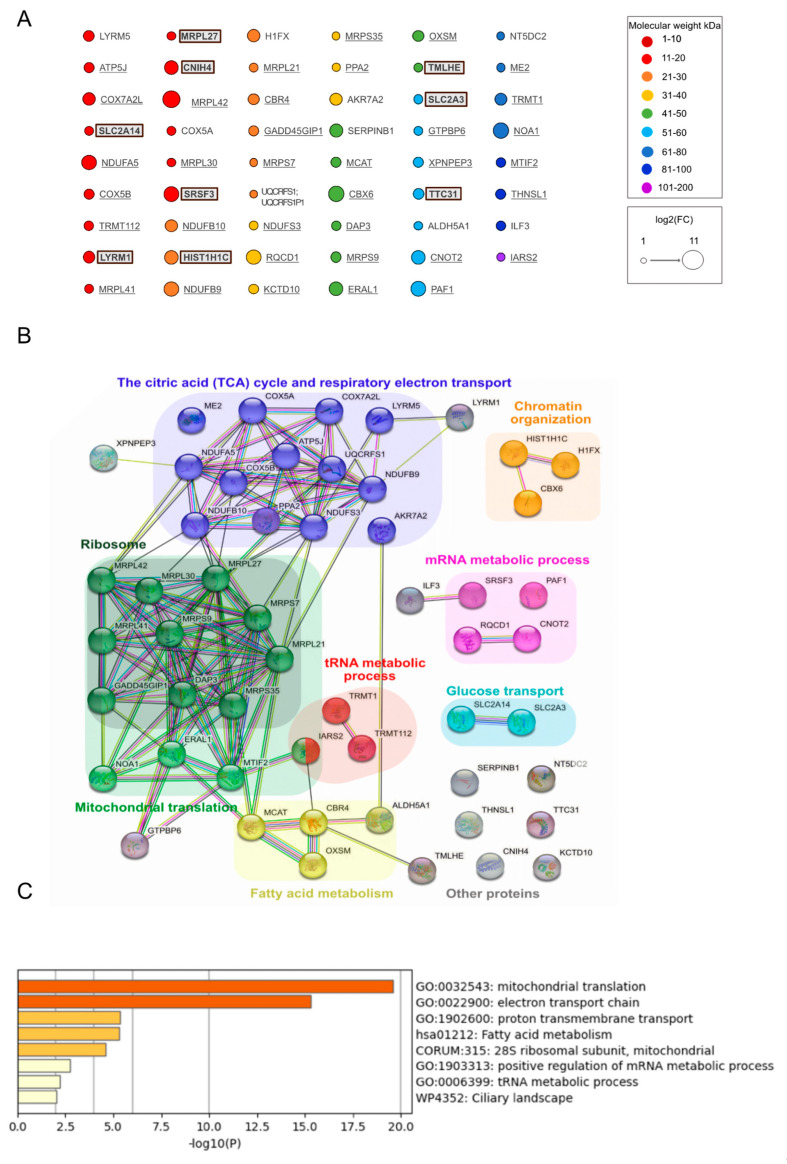
Interactome and functional enrichment analysis of MutIDE. (**A**) 53 enriched proteins, annotated in Figure 3, classified according to molecular weight (circle colors) and fold change (circle diameter); (**B**) Functionally associated protein networks in the MutIDE interactome as identified by STRING analysis; (**C**) Top non-redundant enrichment clusters in the MutIDE interactome, as identified by Metascape. A total of 24% of genes in GO:0032543 and 22% of genes in R-HSA1428517 were enriched. The color scale represents statistical significance levels expressed as −log10.

**Figure 5 biomolecules-13-00890-f005:**
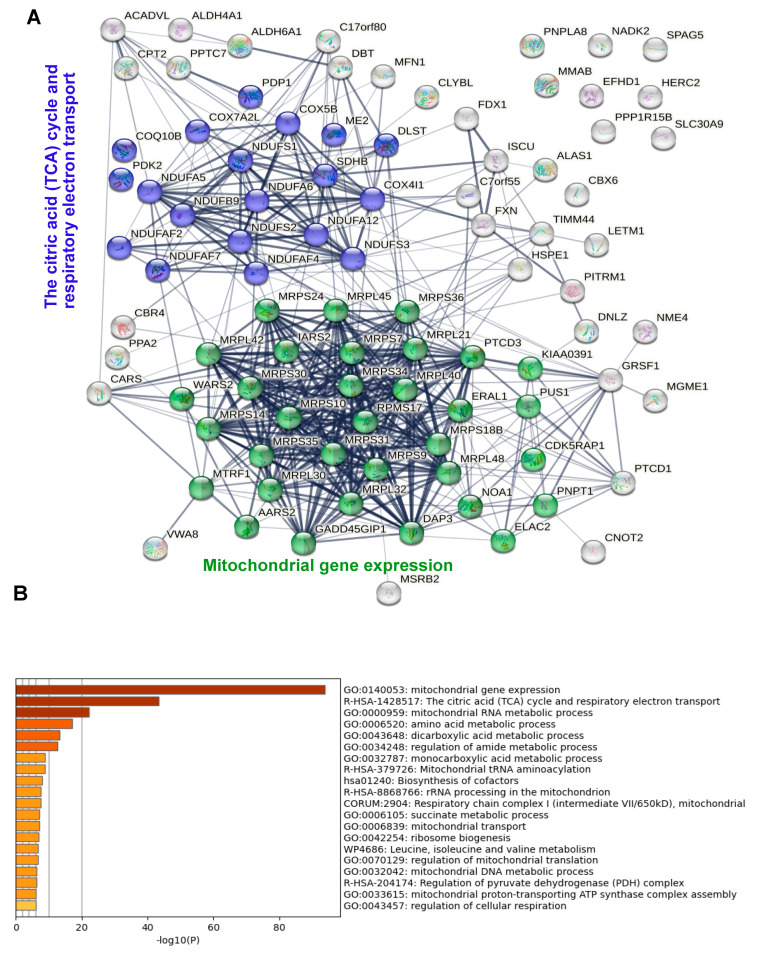
Functional enrichment analysis of the wt IDE interactome obtained by comparing samples with and without biotin. (**A**) Functionally associated protein networks in the IDE interactome as identified by STRING analysis. Proteins belonging to the same biological system are labeled with the identical color; (**B**) Top non-redundant enrichment clusters in the IDE interactome, as identified by Metascape. The color scale represents statistical significance levels expressed as −log10. A total of 21% of genes in GO0140053 and 19% of genes in R-HSA1428517 were enriched.

## Data Availability

The mass spectrometry proteomics data will be available at the Proteome Xchange Consortium.

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
