# Peer review of "Insulin-Degrading Enzyme Interacts with Mitochondrial Ribosomes and Respiratory Chain Proteins"

_biomolecules, 2023, doi:10.3390/biom13060890_

Round 1

Reviewer 1 Report

The manuscript is devoted to the physiological role of mitochondria-targeted insulin-degrading enzyme (IDE). The authors used an exhaustive set of methods to study the localisation sites and prefentually binding proteins of IDE. The results of experiments allow the authors to suppose the role of IDE as a stress-response factor but the hypothesis needs to be tested in the later research. 

The manuscript has no errors nedd to be corrected and can be published in a present form.

Author Response

Thank you very much for your positive comment.

Reviewer 2 Report

The submitted manuscript entitled “Insulin-degrading enzyme targeted to mitochondria: a proteomic study” was dedicated to mitochondrial IDE and its interacting partners and proteolytic substrates. This study scientifically sounds and may be of interest for the journal audience. The manuscript contains 33 references and 9 references published last 5 year. 10 Figures in total and 2 Tables are presented to illustrate the results obtained. However, there are some concerns and recommendations to improve the quality of the manuscript. There are as follows:

1.     I recommend to change the title of the manuscript so it reflects the main results obtained in the study, instead of “proteomic study”.

2.     The same is for the Abstract – more detailed description of the results obtained should be added such as which interacting partners were revealed, which respiratory proteins, etc. Also, in the Abstract the authors stated “do not reveal any proteolytic IDE substrates. We speculate that IDE could adopt similar functions in mitochondria as in the cytosol”. Does it mean that IDE does not exhibit proteolytic activity in the cytosol? If so, where does the enzyme exert its activity? What is Met1?

3.     In the Introduction, the sentence on lines 35-37 has to be supported by References. The sentence on lines 34-35 should be rephrased, because high sequence identity may indicate common evolution, similarity in 3D structure and functions, but not just the indefinite roles in cell biology.

4.     It would be useful to include a short description of structural and functional characteristics of IDE into Introduction section.   

5.     In the Results section, it is not necessary to repeat methods and References should move to the Methods and Discussion sections.

6.     Figure captions on Figs. 3B and 3C, Figs. 4B and 4C, Figs. 5B and 5C are very small and hard to read especially names of proteins in the protein-protein interaction network.

7.     It would be useful to add a Table with the IDE proteolytic substrates list and corresponding kinetic parameters.

English language is qiute good.

Author Response

Thank you very much for your critical and constructive evaluation. We have modified the manuscript accordingly, as described below.

The title has been changed to indicate the main findings.

The abstract has been modified to spell out the main findings, clarify that we refer to functions in addition to proteolysis both in the cytosol and in mitochondria, and removing the Met1 acronym explained later.

In the introduction, a reference has been added concerning IDE substrates, and the interpretation of high sequence identity has been modified as suggested. 

A paragraph outlining structural and functional features of IDE has been added to the introduction.

Several sentences repeating method description in the results section have been eliminated, and the methods section has been modified accordingly where required.

The figure panels 3B,C and 4B,C and 5B have been enlarged as much as possible fitting the page width. I hope the captions are now readable.

I am not sure I understand the comment 7. We did not identify any plausible proteolytic IDE substrates. 

Reviewer 3 Report

I found the paper very interesting and scientifically sound. I think that the aim of the work was challenging and the authors succeded pretty well at giving a first glimpse of the IDE interactome in mitocondria. I just suggest few minor revisions:

1) Title is not very clear. I would change it in "Insulin-degrading enzyme interactome in mitochondria: a proteomic study";

2) In the introduction the authors state that ubiquitin is a substrate of IDE. I would like to point out that such issue is controversial in the literature, so if the authors would like to mention the ubiquitin-IDE relation, they should expand the discussion a little bit and include all the references dealing with such issue (Biochim. Biophys. Acta, 2008,1784, 1122–1126; J. Mol. Biol., 2011, 406, 454–466).

I have found only few typos that could be corrected after a careful proof reading

Author Response

Thank you very much for your positive comments. Please find below the modifications carried out in response to your suggestions.

Modification of the title was also suggested by reviewer 2. I have changed the title to reflect the key findings of the paper.

I have expanded the reference to the interaction between IDE and ubiquitin, adding two more citations including those suggested by you, and spelling out the finding that ubiquitin can inhibit insulin degradation by IDE and not only be degraded by it.

I have verified the complete manuscript for typos and eliminated some, additional errors I might have overlooked can be corrected in the proofs as you suggest.

Round 2

Reviewer 2 Report

The authors properly addressed and replied to all my comments

No comments